# Copper Ionophores as Novel Antiobesity Therapeutics

**DOI:** 10.3390/molecules25214957

**Published:** 2020-10-27

**Authors:** Peter M. Meggyesy, Shashank Masaldan, Sharnel A. S. Clatworthy, Irene Volitakis, Daniel J. Eyckens, Kathryn Aston-Mourney, Michael A. Cater

**Affiliations:** 1Centre for Cellular and Molecular Biology, School of Life and Environmental Sciences, Deakin University, Burwood, Victoria 3125, Australia; pmeggyes@deakin.edu.au (P.M.M.); shashank.masaldan@florey.edu.au (S.M.); sascl@deakin.edu.au (S.A.S.C.); 2Melbourne Dementia Research Centre, The Florey Institute of Neuroscience and Mental Health, Parkville, Victoria 3052, Australia; Irene.volitakis@florey.edu.au; 3Institute for Frontier Materials, Deakin University, Waurn Ponds, Victoria 3216, Australia; deyckens@deakin.edu.au; 4School of Medicine, IMPACT, Institute for Innovation in Physical and Mental Health and Clinical~Translation, Deakin University, Geelong 3220, Australia; k.astonmourney@deakin.edu.au; 5Department of Clinical Pathology, The University of Melbourne, Parkville, Victoria 3010, Australia

**Keywords:** disulfiram, H_2_(gtsm), obesity, copper, ionophore, Antabuse, fat metabolism

## Abstract

The therapeutic utility of the copper ionophore disulfiram was investigated in a diet-induced obesity mouse model (C57BL/6J background), both through administration in feed (0.05 to 1% (*w/w*)) and via oral gavage (150 mg/kg) for up to eight weeks. Mice were monitored for body weight, fat deposition (perigonadal fat pads), metabolic changes (e.g., glucose dyshomeostasis) and pathologies (e.g., hepatic steatosis, hyperglycaemia and hypertriglyceridemia) associated with a high-fat diet. Metal-related pharmacological effects across major organs and serums were investigated using inductively coupled plasma mass spectrometry (ICP-MS). Disulfiram treatments (all modes) augmented hepatic copper in mice, markedly moderated body weight and abolished the deleterious systemic changes associated with a high-fat diet. Likewise, another chemically distinct copper ionophore H_2_(gtsm), administered daily (oral gavage), also augmented hepatic copper and moderated mouse body weight. Postmortem histological examinations of the liver and other major organs, together with serum aminotransferases, supported the reported therapeutic safety of disulfiram. Disulfiram specifically altered systemic copper in mice and altered hepatic copper metabolism, perturbing the incorporation of copper into ceruloplasmin (holo-ceruloplasmin biosynthesis) and subsequently reducing serum copper concentrations. Serum ceruloplasmin represents a biomarker for disulfiram activity. Our results establish copper ionophores as a potential class of antiobesity agents.

## 1. Introduction

Excess body weight is a well-established risk factor for life-threatening diseases, including cardiovascular disease, type 2 diabetes and many types of cancer (e.g., breast, ovarian and prostate). Further, it can complicate and exacerbate chronic conditions such as musculoskeletal disorders (e.g., osteoarthritis) and respiratory diseases (e.g., chronic obstructive pulmonary disease) [1,2]. Behavioural therapies aimed at reducing caloric intake and increasing physical activity are typically ineffective and patients find difficulty adhering to regimes or diets designed for weight loss [3]. Other interventions involve invasive bariatric surgeries [4]. Accordingly, there is a demand for pharmaceutical therapies that can prevent weight gain and/or facilitate weight loss for patients wanting to avoid invasive and often irreversible surgery. 

Studies over the last decade have consistently reported an inverse correlation between body weight and liver (hepatic) copper content [5,6,7,8,9,10,11,12,13,14]. In rodent models of obesity, including ob/ob mice (leptin mutant) [5], and rats fed a high fructose diet [6], copper levels are significantly decreased in hepatic tissue (down ~50%). Likewise, human patients with nonalcoholic fatty liver disease (NAFLD), a common complication of obesity, also present with low hepatic copper (down ~50%) [7]. Conversely, patients with Wilson’s disease, a genetic disorder (*ATP7B* mutation) causing hepatic copper accumulation, have weight loss as a reported symptom [8,9,10,11]. Moreover, murine models of Wilson’s disease have elevated hepatic copper coupled with downregulated hepatic lipid metabolism and decreased body weight [12,13]. Copper deficiency in rats increased the hepatic synthesis of fatty acids with subsequent assembly into triacylglycerols and phospholipids [14], and when sustained (~8 weeks), induced hepatic steatosis (fatty liver) and insulin resistance (IR) [7]. Indeed, copper deficiency has emerged as a putative cause of NAFLD development [7,15]. Lipid metabolism and copper homeostasis are both regulated primarily by the liver [16,17,18]. We here explored the prospect of pharmacologically enhancing hepatic copper to moderate weight gain and preclude the development of associated pathologies. 

Copper-coordinating compounds that can regulate bodily copper have been developed for a wide range of therapeutic indications (e.g., Wilson’s disease, Alzheimer’s disease and cancers) [19,20]. A subclass of these compounds, copper ionophores can elevate intracellular bioavailable copper in target tissues and are being appraised for the treatment of certain cancers [21]. We investigated the copper ionophore disulfiram as a potential antiobesity agent, which has well-characterised pharmacokinetics in mice and humans [22,23] and established tolerance in patients treated for chronic alcoholism (trade name Antabuse) [24,25,26]. Disulfiram, prescribed since the 1950s for the treatment of alcoholism [27], additionally inhibits (copper-independently) the enzyme acetaldehyde dehydrogenase (alcohol metabolism), causing a violent reaction to alcohol consumption (severe hangover symptoms) [28,29,30]. Animal studies (rodents and rabbits) have shown that administration of disulfiram, or its principal metabolite diethyldithiocarbamate (DDTC), elevated copper primarily in the liver [30,31,32,33,34]. Therefore, we sought to determine whether the FDA-approved copper ionophore disulfiram could moderate weight gain in mice.

## 2. Materials and Methods

### 2.1. Animal Experiments

Experiments were conducted in accordance with national and international guidelines and reviewed and approved by the Deakin University Animal Ethics Committee (AEC) (G05-2016). Wild-type (wt) C57BL/6J were purchased from the Australian Animal Resources Centre (Canning Vale, Australia) at 8 weeks of age. Ten-week-old mice were fed either a high-fat diet (HFD), a HFD supplemented with disulfiram (0.05%, 0.125%, 0.25%, 0.5% or 1% *w*/*w*) or a normal chow diet for a maximum of 8 weeks. Alternatively, obesity was induced with 8 weeks of being fed a HFD and then mice were administered copper ionophores in a suspension vehicle (0.9% *w*/*v* NaCl, 0.5% *w*/*v* carboxymethylcellulose, 0.5% *v*/*v* benzyl alcohol, 0.4% *v*/*v* tween-80) via oral gavage (20-gauge bulb-tipped feeding needle) daily for 25 days (150 mg/kg). Mice were maintained at 22 ± 1 °C on a 12 h light/dark cycle with ad libitum access to food and water. Disulfiram (1-(diethylthiocarbamoyldisulfanyl)-*N*,*N*-diethyl-methanethioamide) was purchased from Sigma-Aldrich (Castle Hill, Australia; Cat No. 86720) and H_2_(gtsm) (glyoxalbis(*N*4-methylthiosemicarbazonato)) was synthesised (detailed in the Appendix A). All control and experimental diets were purchased from specialty feeds (Glen Forrest, Western Australia). The normal chow control diet contained 14 kilojoules (kJ)/g with 12% energy from lipids, while the HFD (SF04-001) contained 19 kJ/g with 43% energy from lipids. Diets containing disulfiram were distinguished by food colouring. All animals were weighed twice a week. Food intake was measured weekly and calculated based on the number of mice per cage. Mice were humanely killed by slow-fill CO_2_ and terminal blood samples were collected by cardiac puncture. Blood was allowed to clot on ice for 30 min and serum was collected following two centrifugations at 16,000× *g* for 5 min and then snap frozen. Major organs were snap frozen or preserved for histology as described below (Histology). All harvested organs, including epididymal fat pad, were completely excised and weighed. 

### 2.2. Biochemical Assays

Serum glucose was measured using a FreeStyle Optium Neo Blood Glucose Meter (Abbott Laboratories). Commercially available kits were used to determine serum insulin (ALPCO, Cat No. 80-INSMSU-E01), alanine aminotransferase (ALT) and aspartate aminotransferase (AST) (Abcam, ab105134 and ab105135). The serum was diluted 1:25 for insulin analysis and 1:10 for ALT and AST. Experiments were conducted according to the manufacturer’s instructions.

### 2.3. ICP-MS Analyses

Major organs (liver, kidney, brain, lung, spleen and pancreas) and sera were harvested as described above before having metal concentrations determined using inductively coupled plasma mass spectrometry (ICP-MS) as previously described [20]. Raw ppb values obtained were converted into either µg/g of wet weight for tissues or to µmol/L for serum.

### 2.4. Histology

Major organs (heart, lung, liver, kidney, spleen and brain) were fixed in 10% neutral buffered formalin overnight at 4 °C before being transferred into 70% ethanol and paraffin-embedded. Formalin-fixed, paraffin-embedded (FFPE) tissues were step sectioned (4 µm) and stained with haematoxylin and eosin (H&E) at the Microscopy and Histology Core Facility at the Peter MacCallum Cancer Centre. 

### 2.5. Ceruloplasmin Oxidase Activity

Ceruloplasmin activity in serum was measured spectrophotometrically using *o*-dianisidine dihydrochloride (Sigma-Aldrich, Castle Hill, Australia; Cat No. D3252) as a substrate [35]. Briefly, 5 µL of serum was combined with 75 µL of 0.1 M sodium acetate in two duplicate 96-well plates. Both plates were incubated at 37 °C for 5 min and then 20 µL of 7.88 mM *ο*-dianisidine dihydrochloride was added to each well. Both plates were incubated at 37 °C (one for 5 min and the other for 60 min). The reaction was stopped with 200 µL of 9 M sulfuric acid. Absorbance was read at 540 nm after 5 min using a multiplate reader. Ceruloplasmin oxidase activity was expressed in international units per litre (U/L) using the following formula: (U/L) = (((A_60min_ – A_5min_)/55))/ε × 1/b × 60 × 1000), where A_60min_ and A_5min_ are the absorbances from each respective plate; ε is the molar absorptivity of coloured solutions in terms of substrate consumed (9.6 mL.µmol^−1^.cm^−1^); b = optical length (1 cm); 60 = volume correction factor and 1000 = conversion from mL to L. Sera was taken from all treatment groups (*n* = 14 mice per group) and analysed in duplicate

### 2.6. Western Blot Analyses

To detect ceruloplasmin, mouse serum samples were diluted 1:50 in PBS and fractionated using the Novex Bolt Mini-Gel system (Thermo Fisher, Scoresby, VIC, Australia) on Bolt 4–12% Bis-Tris Plus precast gels. Proteins were then transferred to nitrocellulose membrane (0.45 µm) using the Bolt transfer system and buffer containing 25 mM Tris-HCl, 192 mM glycine and 20% methanol. Membranes were blocked for 90 min at room temperature using 5% (*w*/*v*) skimmed milk powder in TBS-T (10 mM Tris-HCl (pH 8.0), 150 mM NaCl and 0.1% Tween-20). The primary anticeruloplasmin antibody (Dako, Cat No. Q0121) was diluted 1:1000 in TBS-T and membranes were incubated overnight at 4 °C. HRP-conjugated goat-anti-rabbit (Dako, Cat#P0448) antibody was diluted 1:5000 in TBS-T and incubated for 1 h at room temperature. Membranes were developed using ECL reagent (Millipore, Cat No. WBKLS0500, Bayswater, VIC, Australia) and bands visualised on the Gel Doc XR+ system (Bio-Rad, Gladesville, NSW, Australia). Quantification of bands was performed using ImageJ software(https://imagej.net/Downloads).

### 2.7. Synthesis and Validation of H_2_(gtsm)

A round bottom flask was charged with 4-methyl-3-thiosemicarbazide (1.50 g, 10.0 mmol), which was dissolved in EtOH (5 mL) before the addition of glyoxal solution in H_2_O (8.8 M, 5.0 mmol) and subsequent stirring. Glacial acetic acid (5 drops) was added to the mixture prior to refluxing for 5 h. The resulting solution was cooled to 0 °C to allow complete precipitation before filtration and washing with cold H_2_O (2 × 20 mL) followed by cold EtOH (2 × 20 mL). The isolated yellow powder was determined to be the desired product (1.06 g, 91%) in high purity by ^1^H NMR which matched literature reports [36]. ^1^H NMR (DMSO-d_6_, 270 MHz): δ 11.76 (s, 2H), 8.50 (broad s, 2H), 7.72 (s, 2H), 2.96 (d, 6H).

### 2.8. Statistical Analyses

Statistical analyses were performed using one-way ANOVA or two-tailed unpaired *t*-tests where appropriate, calculated on GraphPad PRISM (version 6.0) software. The means of at least triplicate determinations for each test condition were used for comparisons. All data are represented as mean ± SD; probabilities of *p* < 0.05 were considered statistically significant.

## 3. Results

### 3.1. Disulfiram Moderated Weight Gain in Mice

To ascertain whether disulfiram (apo-ligand) can regulate body weight, we treated a diet-induced obesity mouse model (C57BL/6J background) [37] by incorporating disulfiram directly into the feed. Dose tolerance studies determined that disulfiram supplemented at either 0.125% or 0.05% (*w*/*w*) moderated weight gain in mice fed a HFD over eight weeks (Appendix A). Higher concentrations of disulfiram (≥0.25% *w*/*w*) elicited acute weight loss (up to 10%) over a two-week period (Appendix A). Based on these findings, we conducted large-scale studies using a HFD containing 0.05% (*w*/*w*) disulfiram (Figure 1A,B), which markedly limited weight gain across all treated mice (*n* = 14) when compared to the HFD-fed control mice. Mice treated with disulfiram even maintained body weights lower than mice fed normal chow (Figure 1A,B), despite consuming equal kJ/g body weight throughout the feeding regime (Figure 1C). Disulfiram reduced weight gain not by perturbing energy consumption but rather by lowering feed efficiency (weight gain to energy intake ratio) in comparison to control mice (normal chow and HFD fed) (Figure 1D). Data related to body composition and food intake are summarised in Table 1.

Previous toxicity studies in humans and rodents concluded that disulfiram is capable of causing hepatic damage (e.g., focal hepatocellular necrosis), but such complications are rare and typically associated with high doses or extraneous variables (e.g., allergic reactions or pre-existing liver damage) [38,39]. Postmortem histological examination of the liver and other major organs showed no evidence of toxicity in our disulfiram-treated mice (0.05% (*w/w*) in the HFD) (Figure 1E and Appendix A). Mice fed a HFD alone manifested hepatic steatosis (fatty liver) as expected [40], displaying discoloured livers and abundant vacuolar triglyceride accumulation (fatty deposits) (Figure 1E) [41]. Disulfiram-treated mice had no apparent liver pathology, indicating the prevention of HFD-induced hepatic steatosis. Note that weight gain precedes hepatic steatosis manifestation in this obesity mouse model [37]. Mice fed higher quantities of disulfiram (≥1% *w/w*) likewise displayed no liver toxicity (Appendix A), despite these doses eliciting more acute weight loss (Appendix A). Consistent with healthy liver function, mice fed disulfiram (0.05 or 1% *w/w* in the HFD) had serum aminotransferase (AST and ALT) activity levels comparable to control mice (normal chow and HFD fed) (Figure 1F). Together, these results demonstrate that disulfiram when incorporated into the HFD (0.05% *w/w*) moderates weight gain precluding the development of associated liver pathologies.

### 3.2. Disulfiram Prevented Visceral Fat Deposition and Hyperglycaemia in HFD-Fed Mice

Visceral fat deposition in rodents occurs predominantly in perigonadal (epididymal in males), retroperitoneal (kidneys) and mesenteric (alongside the intestinal tract) pads [42]. Perigonadal fat pads are the largest and most accessible and are widely used to study body fat deposition [43,44,45]. Epididymal fat pads were especially evident in mice fed a HFD (Figure 2A). Disulfiram treatment (0.05% (*w/w*) in the HFD) prevented the HFD-associated increase in epididymal fat pads in mice (Figure 2A), consistent with moderating body weight gain through altering fat metabolism (Figure 2A). Hypertriglyceridemia is a common feature of obesity and compounds the risk of obesity-related diseases (e.g., cardiovascular disease, pancreatitis) [46]. Accordingly, serum triglycerides in mice fed a HFD (eight weeks) were significantly higher when compared to mice fed normal chow (Figure 2Bi). Increased serum triglycerides were prevented by disulfiram supplementation (0.05% (*w/w*) in the HFD) (Figure 2Bi).

Insulin resistance (IR) and hyperglycaemia are complications of obesity [47], often leading to the development of type 2 diabetes if untreated [48]. Mice fed a HFD for eight weeks had significantly elevated levels of both insulin and glucose in their serum (Figure 2B(ii,iii)), consistent with the development of a prediabetic or type 2 diabetic state [49]. However, mice treated with disulfiram (0.05% (*w/w*) in the HFD) had serum insulin and glucose concentrations well within normal ranges (Figure 2B). The homeostatic model assessment (HOMA) is a well-documented method for assessing IR [50]. The model compares serum insulin and glucose concentrations to yield an estimate of insulin sensitivity. Consistent with the observed changes to glucose and insulin, mice fed a HFD had a significantly higher HOMA-IR, while disulfiram supplementation abrogated such changes (Figure 2Biv). Collectively, these results demonstrate that disulfiram is an effective antiobesity therapeutic, capable of preventing fat deposition and pathologies (e.g., hyperglycaemia, hypertriglyceridemia, diabetes) associated with a HFD.

### 3.3. Disulfiram Alters Systemic Copper Distribution

We investigated the metal-related pharmacological effects of disulfiram across major organs of treated mice using ICP-MS (Figure 3A). As previously mentioned, the administration of disulfiram or its metabolite diethyldithiocarbamate (DDTC) elevates copper primarily in the liver and to a lesser extent the brain [30,31,32,33,34]. Accordingly, mice treated with disulfiram (0.05% (*w*/*w*) in the HFD) for eight weeks had significantly elevated liver (~8.2-fold) and brain (~2.8-fold) copper levels in comparison to control mice (normal chow and HFD fed) (Figure 3A). Disulfiram treatment also slightly elevated copper levels in the pancreas (~1.2-fold) and spleen (~1.3-fold), while having a negligible impact upon other investigated organs (lung and kidneys) (Figure 3A). Analyses of other organ metal levels, including iron and zinc, revealed no significant aberrations with disulfiram treatment (Appendix A). Hepatic steatosis is known to cause hepatic iron dyshomeostasis [51], and consistently, mice fed a HFD had slightly elevated hepatic iron content (Appendix A). Disulfiram supplementation mitigated HFD-induced iron elevation (Appendix A).

The liver mediates systemic copper homeostasis, sequestering newly absorbed dietary copper and regulating copper incorporation into both serum components (systemic distribution) and bile (excretion) [52,53]. Elevated hepatic copper is generally coupled with elevated serum copper [54,55,56]; however, mice treated with disulfiram (0.05% (*w*/*w*) in the HFD) displayed a paradoxical decrease in serum copper concentration (~1.4-fold) (Figure 3B). To reconcile this discrepancy, we measured the activity and expression of the principal copper-containing serum protein, ceruloplasmin (Figure 3C,D). Ceruloplasmin coordinates copper (six atoms) during its hepatic biosynthesis and when secreted accounts for more than 70% of circulatory copper [21]. The copper-dependent activity of ceruloplasmin was significantly reduced in mice fed a HFD with disulfiram (0.05% (*w*/*w*)) (Figure 3C), consistent with there being decreased serum copper concentrations (Figure 3B) [57,58]. Western blot analyses revealed that mice fed disulfiram secreted less holo-ceruloplasmin (copper-bound) into their serum but maintained appreciable levels of apo-ceruloplasmin (copper-free) (Figure 3D). Reduction in the holo- versus apo-ceruloplasmin ratio (Figure 3D) indicates altered hepatic copper metabolism, whereby less copper is incorporated during ceruloplasmin biosynthesis [59]. These results demonstrated that disulfiram, despite elevating hepatic copper levels (Figure 3A), markedly altered hepatic copper distribution, reducing holo-ceruloplasmin biosynthesis and serum copper levels. 

### 3.4. Copper Ionophores Moderate Weight Gain When Administered via Oral Gavage

We investigated whether disulfiram administered as a daily bolus dose, as opposed to direct incorporation into feed (Figure 1A), was sufficient to moderate weight gain in obese mice (Figure 4). Mice fed a HFD for eight weeks were subsequently coadministered disulfiram (150 mg/kg) via oral gavage for a further 25 days. Previous studies administered disulfiram (ligand alone and with copper) at double our daily dosage (300 mg/kg) and reported no adverse side effects [60]. Disulfiram administered daily significantly attenuated weight gain when compared to mice fed a HFD diet with vehicle control (orally gavaged) (Figure 4A). We further investigated a second copper ionophore, H_2_(gtsm), that harbours distinct chemistry (bis(thiosemicarbazone) family) and pharmacokinetics from disulfiram [27]. H_2_(gtsm) is not metabolised in vivo (disulfiram is reduced to diethyldithiocarbamate (DDTC)), does not inhibit hepatic acetaldehyde dehydrogenase (antialcoholism activity) and coordinates copper in a 1:1 ratio [61,62,63]. Despite these differences, both disulfiram and H_2_(gtsm) dissociate coordinated copper under the reductive intracellular environments, redistributing copper into a bioavailable pool [20,27,61]. A cohort of mice were coadministered H_2_(gtsm) (apo-ligand) (150 mg/kg) with a HFD under the same conditions used above for disulfiram (Figure 4A). The dose of H_2_(gtsm) used was previously administered to mice without adverse side effects and weight loss was not reported [64,65]. Analogous with disulfiram treatment, H_2_(gtsm) administered once daily with a HFD markedly reduced weight gain in mice (Figure 4A). Neither disulfiram nor H_2_(gtsm) when administered orally altered food intake (energy consumption) (not shown), but both treatments significantly reduced feeding efficiency (weight gain to energy intake ratio) (Figure 4B).

Concluding the treatment regime, organs and sera were harvested from mice (1 h after final dose) and analysed for metal-related pharmacological effects by ICP-MS (Figure 4C). Expectedly, mice orally gavaged with either disulfiram or H_2_(gtsm) presented with increased liver (1.6–4.9-fold) and brain (1.2–1.5-fold) copper levels (Figure 4C(i,ii)) [32,33,34]. Disulfiram treatment also elevated copper levels in the pancreas (~1.8-fold) (Figure 4D). Other investigated organs (lungs and kidneys) and sera had copper levels within their normal respective range (Figure 4C(v,vi)). The variability in copper accumulation within a given organ (e.g., liver) between treatments (disulfiram versus H_2_(gtsm)) can be explained, in part, due to differences in pharmacokinetics at the time of collection. Nonetheless, disulfiram modulated systemic copper analogously when administered by oral gavage (Figure 4) or through feed (Figure 1). Together, these results demonstrated that chemically distinct copper ionophores administered to mice as a daily bolus dose modulated liver copper levels and moderated weight gain in mice fed a HFD. 

## 4. Discussion

Elevating liver copper through dietary supplementation is impractical due to the organs’ proficiency to eliminate excess copper [21]. We demonstrated that pharmacological augmentation of hepatic copper with copper ionophores is associated with lower body weights (Figure 1 and Figure 4) and reduced fat deposition. Disulfiram treatment further prevented metabolic changes and pathologies (e.g., hyperglycaemia, hypertriglyceridemia) associated with a HFD (Figure 1 and Figure 2). Excess hepatic copper is countered by the copper-transporter Atp7b trafficking to post-Golgi excretory vesicles (pericanalicular/apical), which expels surplus copper into bile [66]. Atp7b further mediates copper incorporation into ceruloplasmin when residing at the *trans*-Golgi network under basal (regular) copper conditions [67]. Disulfiram treatment in feed paradoxically caused elevated hepatic copper coupled with reduced holo-ceruloplasmin biosynthesis (metalation) and reduced sera copper concentrations (Figure 3 and Figure 4). Conceivably, a post-Golgi steady-state localisation of Atp7b, perpetuated by continuous disulfiram treatment, would impede copper translocation into the hepatic secretory pathway and therefore ceruloplasmin metalation. However, we cannot rule out the possibility that disulfiram directly inhibited Atp7b or delivered copper to an alternative hepatic pathway/compartment(s). Nonetheless, copper incorporation into ceruloplasmin is perturbed following disulfiram treatment (Figure 3D), despite elevating hepatic copper, and treated mice displayed a systemic copper profile analogous to when *Atp7b* is mutated [20].

Disulfiram is rapidly absorbed from the gastrointestinal tract (up to 99%) unlike other copper ionophores (e.g., clioquinol), avoiding changes to digestive functions [22,23]. Despite disulfiram being known to primarily target the liver [32,33] extrahepatic copper-dependent effects could also potentially contribute to modulating body weight. For instance, copper was recently identified as a mediator of cyclic-AMP-dependent lipolysis (hydrolysis of triglycerides to release fatty acids) in a cell culture model of adipocytes (3T3-L1) [68]. We also confirmed that disulfiram elevated brain copper content (Figure 3A) [30,31] and precluded the development of obesity-associated brain atrophy in mice fed a HFD (Table 1), which is reportedly associated with cognitive decline [69]. Copper-dependent signalling pathways in the brain facilitate neurotransmission and promote neuronal growth (neurogenesis) [70,71] and may conceivably support the brain’s role in regulating energy balance and body weight [72]. 

Disulfiram has long been administered to humans as a daily oral bolus dose (e.g., 500 mg) for the treatment of alcoholism [25,73], which could also represent the treatment paradigm for the management of body weight (Figure 4). However, large trials in humans have not reported weight loss in response to disulfiram treatment for alcoholism [74,75]. In fact, a meta-analysis of large-scale trials conducted over the last 20 years did not report any effects on body weight [76]. High frequency of nonadherence, combined with gastrointestinal symptoms (disulfiram-alcohol-induced vomiting, diarrhoea and lack of appetite), fluctuate body weight in alcoholic patients and thus weight change measurements cannot be reliably interpreted. While human patients have tolerated high bolus doses of disulfiram (≥1.5 g daily) with minimal adverse side effects [77], the minimal dose for therapeutic utility in the context of weight management will need to be determined. Encouragingly, disulfiram incorporated into feed was an effective route of administration for modulating weight in mice (Figure 1), which indicates that smaller doses administered concurrently with meals may prove effective in humans. Multiple daily oral doses may also counteract the short pharmacokinetic half-life of disulfiram (~8 h) [22]. 

We previously demonstrated that the copper ionophore clioquinol induces weight loss in several mouse models [61]. Further, disulfiram has been recently described to normalise bodyweight in a mouse model of obesity [78]. In this study, we validated the link between copper and lipid metabolism by demonstrating that two chemically distinct copper ionophores, disulfiram and H_2_(gtsm), can modulate body weight in a diet-induced obesity mouse model. Further, our results establish the FDA-approved copper ionophore disulfiram as an attractive candidate for pharmaceutical repurposing as an antiobesity agent. 

## Figures and Tables

**Figure 1 molecules-25-04957-f001:**
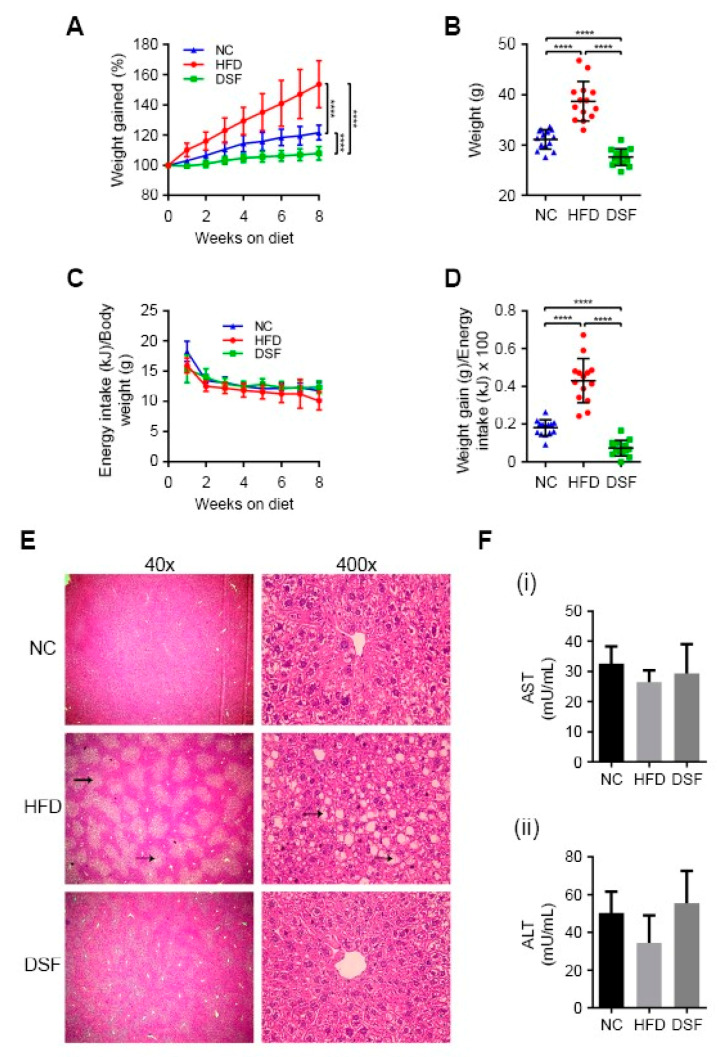
Disulfiram moderated weight gain in mice fed a high-fat diet (HFD). (**A**) Mice fed a HFD supplemented with disulfiram (0.05% *w*/*w*) (DSF) (*n* = 14) progressively gained less weight than mice fed either a HFD alone (HFD) (*n* = 14) or normal chow (NC) (*n* = 14). Weight change as a percentage over the 8-week feeding regime is shown. (**B**) Endpoint weights of mice fed either normal chow (NC) (*n* = 14), a HFD (HFD) (*n* = 14) or a HFD supplemented with disulfiram (0.05% *w*/*w*) (DSF) (*n* = 14) after 8 weeks. (**C**) Energy consumption was comparable between mice fed either normal chow (NC) (*n* = 14), a HFD (HFD) (*n* = 14) or a HFD supplemented with disulfiram (0.05% *w*/*w*) (DSF) (*n* = 14) over 8 weeks. Results were expressed as kilojoules and were normalised against individual mouse body weights (per gram). (**D**) Mice fed a HFD supplemented with disulfiram (0.05% *w*/*w*) (DSF) (*n* = 14) had significantly decreased feeding efficiencies in comparison to mice fed either normal chow (NC) (*n* = 14) or a HFD (HFD) alone (*n* = 14). Feeding efficiency represents the ratio of weight gain to energy (kilojoules) intake. (**E**) Disulfiram supplementation into the HFD (0.05% *w*/*w*) prevented HFD-induced hepatic steatosis. Representative H&E stained sections of liver from mice fed either normal chow (NC) (*n* = 4), a HFD (HFD) (*n* = 4) or a HFD supplemented with disulfiram (0.05% *w*/*w*) (DSF) (*n* = 4) after 8 weeks. Fatty deposits in mice fed a HFD (black arrows) and magnifications are shown. (**F**) Mice fed a HFD supplemented with disulfiram (0.05% *w*/*w*) (DSF) (*n* = 5) had serum aminotransferase (aspartate transaminase (AST) and alanine transaminase (ALT)) activity levels comparable to mice fed either normal chow (NC) (*n* = 5) or a HFD (HFD) alone (*n* = 5). Serum aminotransferase activities are expressed as milliunits per mL. Results represent mean ± SD. (**** *p* < 0.0001).

**Figure 2 molecules-25-04957-f002:**
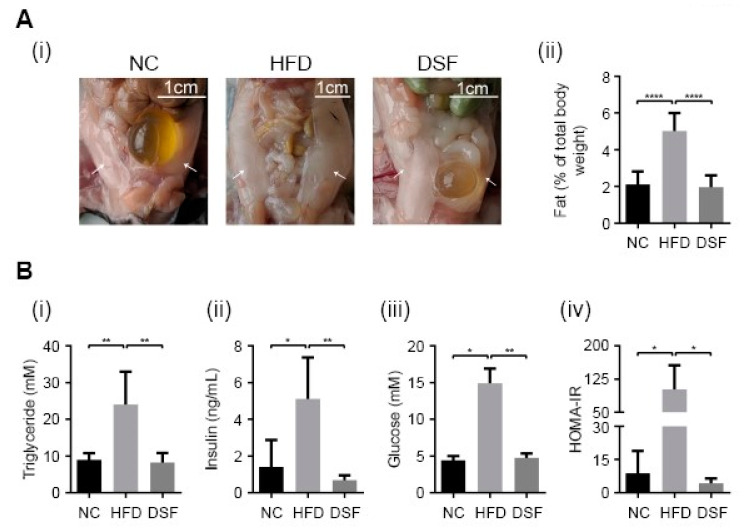
Disulfiram prevented fat deposition and pathologies associated with a high-fat diet (HFD)**.** (**A**) (i) Disulfiram supplementation into the HFD (0.05% *w/w*) prevented bilateral epididymal fat pad development in mice. Representative photographs of intact epididymal fat pads in mice fed either normal chow (NC) (*n* = 14), a HFD (HFD) (*n* = 14) or a HFD supplemented with disulfiram (0.05% *w*/*w*) (DSF) (*n* = 14) after 8 weeks. Epididymal fat pads in mice fed a HFD (white arrows) and the scale is shown. (ii) Endpoint weights of intact epididymal fat pads dissected from mice fed either normal chow (NC) (*n* = 14), a HFD (HFD) (*n* = 14) or a HFD supplemented with disulfiram (0.05% *w*/*w*) (DSF) (*n* = 14) after 8 weeks. Fat pads were weighed and expressed as a percentage of individual mouse body weight. (**B**) Disulfiram supplementation (0.05% *w*/*w*) prevented metabolic changes and pathologies associated with a HFD. Mice fed a HFD supplemented with disulfiram (0.05% *w*/*w*) (DSF) (*n* = 5) had normal levels of (i) serum triglycerides, (ii) insulin, (iii) glucose and (iv) insulin resistance (HOMA-IR), when compared to mice fed with either normal chow (NC) (*n* = 5) or a HFD (HFD) (*n* = 5). Hypertriglyceridemia, hyperglycaemia and insulin resistance (diabetic state) can be seen in mice fed a HFD. Results represent mean ± SD. (* *p* < 0.05; ** *p* < 0.01; **** *p* < 0.0001).

**Figure 3 molecules-25-04957-f003:**
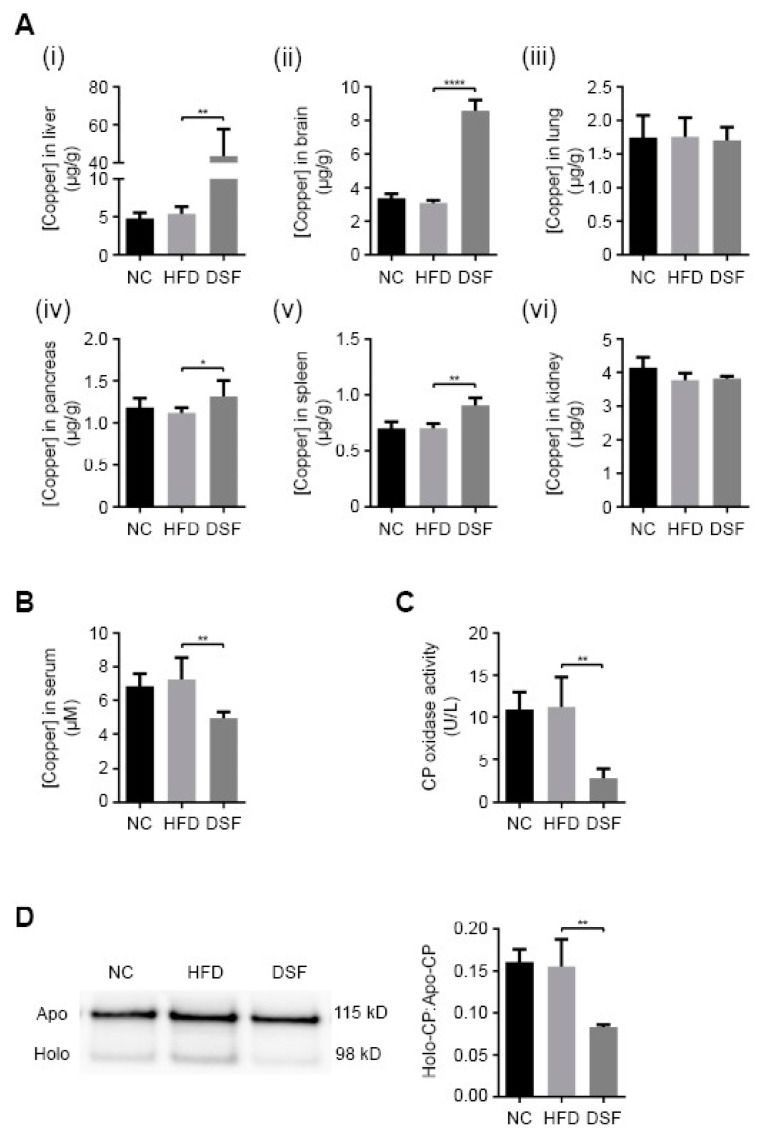
Disulfiram alters systemic copper distribution in mice. (**A**) Inductively coupled plasma mass spectrometry (ICP-MS) was used to measure copper concentrations in (i) liver, (ii) brain, (iii) lung, (iv) pancreas, (v) spleen and (vi) kidney of mice fed either normal chow (NC) (*n* = 5), a HFD (HFD) (*n* = 5) or a HFD supplemented with disulfiram (0.05% *w*/*w*) (DSF) (*n* = 5) after 8 weeks. Results represent mean ± SD and are shown as µg/g wet weight. (**B**) ICP-MS was used to measure serum copper concentrations in mice fed either normal chow (NC) (*n* = 5), a HFD (HFD) (*n* = 5) or a HFD supplemented with disulfiram (0.05% *w*/*w*) (DSF) (*n* = 5) after 8 weeks. Results represent mean ± SD and are shown as µM. (**C**,**D**) Disulfiram reduces holo-ceruloplasmin biosynthesis and secretion. (**C**) Ceruloplasmin oxidase activity was measured using the o-dianisidine dihydrochloride-based assay in sera from mice fed either normal chow (NC) (*n* = 14), a HFD (HFD) (*n* = 14) or a HFD supplemented with disulfiram (0.05% *w/w*) (DSF) (*n* = 14) after 8 weeks. Results are expressed as unit/litre (U/L). (**D**) Western blot analyses of apo- and holo-ceruloplasmin secretion into sera of mice fed either normal chow (NC) (*n* = 4), a HFD (HFD) (*n* = 4) or a HFD supplemented with disulfiram (0.05% *w*/*w*) (DSF) (*n* = 4) after 8 weeks. Ceruloplasmin was detected as two bands; apo-ceruloplasmin (115 kDa; copper-free) and holo-ceruloplasmin (98 kDa; copper-bound). Densitometry analysis was conducted to determine the ratio of holo- versus apo-ceruloplasmin in sera of mice fed either normal chow (NC) (*n* = 4), a HFD (HFD) (*n* = 4) or a HFD supplemented with disulfiram (0.05% *w*/*w*) (DSF) (*n* = 4) after 8 weeks. (* *p* < 0.05; ** *p* < 0.01; **** *p* < 0.0001).

**Figure 4 molecules-25-04957-f004:**
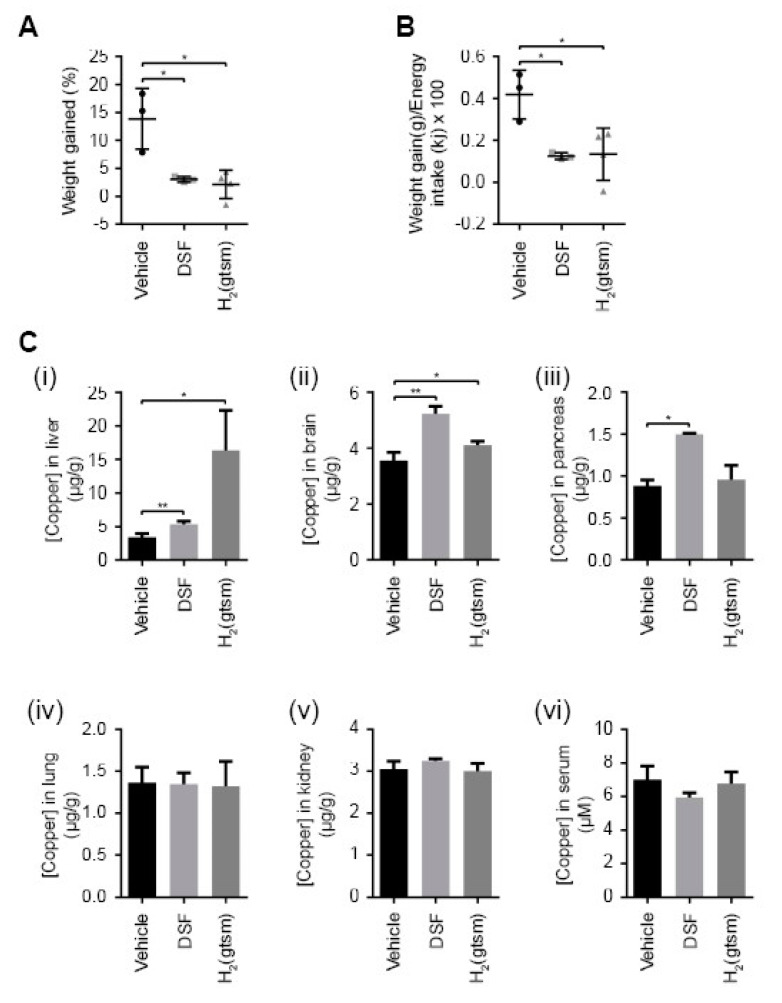
Copper ionophores disulfiram and H_2_(gtsm) moderated weight gain when administered via oral gavage. (**A**) Mice fed a HFD for 8 weeks were subsequently cotreated daily with either disulfiram (150 mg/kg) (DSF) (*n* = 3), H_2_(gtsm) (150 mg/kg) (*n* = 4) or vehicle control (Vehicle) (*n* = 3) via oral gavage. Weight change as a percentage after 25 days of treatment is shown. (**B**) Mice treated with disulfiram (150 mg/kg) (DSF) (*n* = 3) and H_2_(gtsm) (150 mg/kg) (*n* = 4) had significantly decreased feeding efficiencies in comparison to the vehicle control mice (*n* = 3). Feeding efficiency represents the ratio of weight gain to energy (kilojoules) intake. (**C**) ICP-MS was used to measure copper concentrations in (i) liver, (ii) brain, (iii) pancreas, (iv) lung, (v) kidney and (vi) serum of mice treated with either disulfiram (150 mg/kg) (DSF) (*n* = 3), H_2_(gtsm) (150 mg/kg) (*n* = 4) or vehicle control (Vehicle) (*n* = 3) after 25 days of oral gavage. Results represent mean ± SD and are shown as µg/g wet weight for tissues and µM for serum. (* *p* < 0.05; ** *p* < 0.01).

**Table 1 molecules-25-04957-t001:** Body composition, food and nutrient intakes in mice fed normal chow (NC), a high-fat diet (HFD) or a high-fat diet with 0.05% *w*/*w* disulfiram (DSF).

Body Weight and Food Intake Data	Groups			HFD vs. DSF
	NC	HFD	DSF	*p*
Final body weight (g)	31.1 ± 1.9	38.7 ± 3.9	27.6 ± 1.6	<0.0001
Body weight gain (%)	21.5 ± 4.9	53.7 ± 15.5	7.9 ± 4.4	<0.0001
Liver weight ^1^	4.7 ± 0.3	3.7 ± 0.5	3.7 ± 0.2	ns
Brain weight ^1^	1.4 ± 0.1	1.1 ± 0.1	1.5 ± 0.04	<0.0001
Pancreas weight ^1^	0.61 ± 0.09	0.60 ± 0.07	0.63 ± 0.05	ns
Kidney weight ^1^	0.65 ± 0.05	0.51 ± 0.04	0.55 ± 0.05	ns
Lung weight ^1^	0.35 ± 0.04	0.31 ± 0.07	0.34 ± 0.04	ns
Spleen weight ^1^	0.23 ± 0.03	0.25 ± 0.02	0.27 ± 0.03	ns
Food intake (g) ^2^	26.2 ± 3.2	21.5 ± 1.7	18.4 ± 2.0	<0.0001
Relative food intake ^3^	0.94 ± 0.05	0.64 ± 0.06	0.69 ± 0.05	0.02
Energy intake (kJ) ^2^	366.4 ± 44.2	407.5 ± 32.3	349.4 ± 37.5	<0.0001
Relative energy intake ^3^	13.2 ± 0.7	12.1 ± 1.2	13.1 ± 0.9	0.02
Lipid intake (kJ) ^2^	43.9 ± 5.3	175.2 ± 13.9	150.2 ± 16.1	<0.0001
Protein intake (kJ) ^2^	84.3 ± 10.2	85.6 ± 6.8	73.4 ± 7.9	<0.0001

^1^ Values expressed as a percentage of total body weight. ^2^ Amounts expressed/mouse/week. ^3^ Values are g/kJ consumed/gram of body weight. Values are mean ± SD (*n* = 14 mice/group).

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
