# Peer review of "Copper Ionophores as Novel Antiobesity Therapeutics"

_molecules, 2020, doi:10.3390/molecules25214957_

Round 1

Reviewer 1 Report

Manuscript #Molecules_950209

The manuscript by Peter M. Meggyesy et al. reports on the application of copper-ionophores to the prevention of obesity. Obesity is currently a risk factor with implication in several diseases, that have a high economic impact. Thus, the identification of therapeutic agents that have the potential to reduce it, without surgical procedures is of outmost importance, due to the eating behavior of today society. This study has the advantage that are testing a FDA-approved drug for the therapeutic of obesity.

The manuscript is well written and the figures have been adequately chosen.

Questions

  1. The authors mention that there is a change in the level of copper incorporated into ceruloplasmin. This will have a direct effect in the iron-metabolism and iron homeostasis. Did the authors look for these changes? What were the observation in the level of iron in the blood, liver (such as, haemoglobin, ferritin)
  2. Line 56-57 , revise font size.
  3. It was not clear to me, whether the dosage administered was identical to the one that is approved for the treatment of alcoholism.
  4. The dosage and period that the mice were subjected to this treatment is similar to what would be administered to humans? What were the systemic changes monitored?

Author Response

Questions

1. The authors mention that there is a change in the level of copper incorporated into ceruloplasmin. This will have a direct effect in the iron-metabolism and iron homeostasis. Did the authors look for these changes? What were the observation in the level of iron in the blood, liver (such as, haemoglobin, ferritin)

While we did not measure indicators of iron homeostasis such as ferritin, Hb or Tf saturation, in the serum, our ICPMS analyses of various organs and sera provided us with total iron levels. Iron levels did not change significantly enough to account for the biological effect of ionophores on weight gain. The change in iron levels in major organs and serum are included in supplementary Figure 4B.

2. Line 56-57 , revise font size.

We have made the change and corrected other similar errors in the manuscript.

3. It was not clear to me, whether the dosage administered was identical to the one that is approved for the treatment of alcoholism.

The doses we administered via oral gavage 150mg/kg are considered safe for use in rodents but are higher than those typically used for treatment of alcoholism in humans (500mg/day). While direct comparisons between mice and humans are difficult to make, nevertheless, the doses we administered in feed were much lower. It is important to note that DSF is well tolerated at even higher doses in humans >1.5g/ day (as mentioned in manuscript).

4. The dosage and period that the mice were subjected to this treatment is similar to what would be administered to humans? What were the systemic changes monitored?

We covered the possible modes of DSF treatment in humans in the discussion. Weight gain in mice was monitored during treatment periods, and post-mortem we monitored many different systemic physiological and pathological outputs such as bilateral epididymal fat pad development, insulin resistance, insulin and blood glucose levels, liver function, etc.

Reviewer 2 Report

Referee report on the manuscript “ Copper-ionophores as novel anti-obesity therapeutics” by Peter M. Meggyesy  et al. to be published in Molecules (manuscript ID: Molecules-950209).

General comments

The manuscript deals with an interesting and important topic. There may be significant interest in the use of copper chelators as anti-obesity agents. One of the tested active ingredients (disulfiram) has long been known and can be used in the clinic. A number of conceptual issues arise in relation to true applicability of he presented drugs as anti-obesity agents, these are already beyond the scope of this research.

The MS is properly written and its high quality meets the requirements of Molecules.

Although this is a well written manuscript, I suggest some corrections:

  1. Since Molecules a chemical journal, the formulas of the chelators should be included in the MS. Moreover H2(gtsm) is not a chemical name, and glyoxalbis(N4-methylthiosemicarbazonato) (line 87) looks not better. I can only guess, maybe this is a glyoxal bis[N(4)-methylthio semicarbazone, but should be clarified.

  1. The affinity of the applied chelators (all stability data is missing) used for copper and other important metals (e.g. Zn) is very different, which should be mentioned.

  1. The dose used for disulfiram is not clear. It is written: “HFD supplemented with disulfiram (0.05%, 0.125%, 0.25%, 0.5% or 1% w/w)”. The same problem in many figures, 0.05% w/w (mass percent) can be found. What does it mean? What is the dose in mg/kg? Please indicate the dose in mg/kg everywhere.

All suggested modifications fall in the category of a "minor revision".

Author Response

 1. Since Molecules a chemical journal, the formulas of the chelators should be included in the MS. Moreover H2(gtsm) is not a chemical name, and glyoxalbis(N4-methylthiosemicarbazonato) (line 87) looks not better. I can only guess, maybe this is a glyoxal bis[N(4)-methylthio semicarbazone, but should be clarified.

As we did not describe any of the chemistry or structure activity relationship involved, we did not consider inclusion of the chemical structures, however these have been previously reported numerous times as we referenced (22, 23). We have only used well characterised copper ionophores. To clarify, the chelator is called glyoxalbis (N4-methylthiosemicarbazonato) as can be seen in prior publications (e.g. Cater, 2013 ACS Chemical Biol).

2. The affinity of the applied chelators (all stability data is missing) used for copper and other important metals (e.g. Zn) is very different, which should be mentioned.

As our manuscript details the biological activity of DSF and CuII(gtsm) in an in vivo setting we find the description of affinity constants, typically measured under defined conditions, outside and not relevant to the scope of our work. However, both ionophores used have been shown to preferentially bind copper and as can be seen by our metal analyses of organs and serum, only affects copper in vivo.

3. The dose used for disulfiram is not clear. It is written: “HFD supplemented with disulfiram (0.05%, 0.125%, 0.25%, 0.5% or 1% w/w)”. The same problem in many figures, 0.05% w/w (mass percent) can be found. What does it mean? What is the dose in mg/kg? Please indicate the dose in mg/kg everywhere.

The description of drug supplemented feed as weight percentage is a standard convention (1%w/w = 10g/kg); mice feed ad libitum. This is to differentiate it from the mg/kg doses as mentioned for oral gavage, which is the amount of drug per kg weight of the animal it is administered to. 

Reviewer 3 Report

In this manuscript, Meggyesy et al. Suggest disulfiram and other copper-ionophores as possible anti-obesity drugs. Based on in vivo mouse experiments, the authors demonstrated that disulfiram suppressed high-fat induced obesity in the mouse model and associated markers (fat deposition, glucose metabolism, liver pathology). The anti-obesity properties of disulfiram were already reported previously (Bernier et al., Cell Metabolism, 2020). What is novel in this manuscript, is the proposed mechanism of action of disulfiram involving altered copper homeostasis, further supported by the similar effect of chemically distinct copper-binding compound (H2(gtsm)- validated copper tracer), which also lowered high-fat induced obesity. However, the mechanism of action involving copper homeostasis is not entirely supported by the data presented in the manuscript and should be formulated with more precautions (see the points below).

  1. The described adverse effect of disulfiram in human patients is a loss of appetite and a metallic taste in the mouth. Food intake and energy intake were both significantly lower in the HFD+DSF group than the HFD group alone, according to table 1. How were these results taken into account? Cannot be this the most significant factor for the observed effect on weight gain? The authors should at least discuss it.

  1. The main novelty of the manuscript is the suggested mechanism of action of disulfiram (or other copper-chelating agents) involving altered copper homeostasis. Indeed, dramatically increased copper levels (8.2 fold) were observed in the liver of animals on a diet with disulfiram. Decreased copper levels in serum were also detected. But, when disulfiram was applied by direct oral gavage, a minimal effect on copper metabolism was observed (in the liver, it was only 1.6 fold increase of copper, no change in copper content in serum), despite the very high dose of DSF used (150 mg/kg/day). Yet, both treatment approaches (disulfiram mixed with feed or via oral gavage) have a similar effect on body weight gain. How is this possible if the mechanism of action should involve altered copper metabolism? Such data contradict such a claim. The authors speculate that disulfiram applied via oral gavage could have different metabolism, which is very likely because in the first case, it was mixed with a high-fat diet, but in the second case, it was not resuspended in oil (a common practice in DSF field) but instead in a mixture of pharmacological excipients. But if different metabolism of disulfiram led to a negligible effect on copper metabolism, why the impact on body weight was preserved (if altered copper metabolism should be the mechanism of action)? In this discrepancy, I see the weakest point of the manuscript. The authors should face it properly and admit that in DSF treated mice, the effect on copper heomeostasis does not correlate well with the anti-obesity effect of DSF.

  1. Authors call disulfiram as copper ionophore and claim that it increases the levels of bioavailable copper. Such an effect of disulfiram was, however, never confirmed. Data presented in the manuscript instead indicate the opposite. Despite the accumulation of copper in the liver, circulating levels of copper were decreased after disulfiram. Similarly, ceruloplasmin containing copper was also reduced. Such a claim should be reformulated or supported by more persuasive data.

  1. The authors should at least discuss the previous work demonstrating the anti-obesity effect of disulfiram (Bernier et al., Cell Metabolism, 2020, doi.org/10.1016/j.cmet.2020.04.019), as this paper used very similar methodology, experiments, and readouts, came to a similar conclusion regarding the effect of disulfiram on weight, but without any explanation.

Author Response

1. The described adverse effect of disulfiram in human patients is a loss of appetite and a metallic taste in the mouth. Food intake and energy intake were both significantly lower in the HFD+DSF group than the HFD group alone, according to table 1. How were these results taken into account? Cannot be this the most significant factor for the observed effect on weight gain? The authors should at least discuss it.

The relative caloric intake did not change but in fact HFD mice on DSF consumed slightly higher caloric intake per gram bodyweight than those on HFD alone. Therefore, there was no reduced palatability as mice with DSF in the feed ate more. We have made this assertion and discussed this in lines 170-174 in the manuscript:

“Mice treated with disulfiram even maintained body weights lower than mice fed normal chow (Fig 1A&B), despite consuming equal kJ/g body weight throughout the feeding regime (Fig 1C). Disulfiram therefore reduced weight gain not by perturbing energy consumption, but rather by lowering feed efficiency (weight gain to energy intake ratio) in comparison to control mice (normal chow and HFD fed)”

2. The main novelty of the manuscript is the suggested mechanism of action of disulfiram (or other copper-chelating agents) involving altered copper homeostasis. Indeed, dramatically increased copper levels (8.2 fold) were observed in the liver of animals on a diet with disulfiram. Decreased copper levels in serum were also detected. But, when disulfiram was applied by direct oral gavage, a minimal effect on copper metabolism was observed (in the liver, it was only 1.6 fold increase of copper, no change in copper content in serum), despite the very high dose of DSF used (150 mg/kg/day). Yet, both treatment approaches (disulfiram mixed with feed or via oral gavage) have a similar effect on body weight gain. How is this possible if the mechanism of action should involve altered copper metabolism? Such data contradict such a claim. The authors speculate that disulfiram applied via oral gavage could have different metabolism, which is very likely because in the first case, it was mixed with a high-fat diet, but in the second case, it was not resuspended in oil (a common practice in DSF field) but instead in a mixture of pharmacological excipients. But if different metabolism of disulfiram led to a negligible effect on copper metabolism, why the impact on body weight was preserved (if altered copper metabolism should be the mechanism of action)? In this discrepancy, I see the weakest point of the manuscript. The authors should face it properly and admit that in DSF treated mice, the effect on copper heomeostasis does not correlate well with the anti-obesity effect of DSF.

The increase in copper because of a bolus dose of DSF are expectedly lower when compared to a constant administration of DSF via feed. The discrepancy is due to the terminal nature of measurement by ICPMS, which does not capture the highest transient increase in copper following the administration of a bolus dose of DSF. Establishing the highest liver copper achieved following bolus dose of DSF would require multiple biopsies at many timepoints, to capture the transient increase in copper more accurately, but does not ultimately provide any additional information regarding the benefit of DSF and is this hard to justify ethically. Nonetheless, even a 1.6- fold increase of copper is significant as copper is tightly controlled in hepatocytes. Furthermore, DSF given via feed or via bolus dose produced comparable copper metabolic effects, as detailed in the results. We certainly did not ‘speculate that disulfiram applied via oral gavage could have a different mechanism’ as insinuated by this Reviewer. What these results clearly demonstrate is that continual administration of DSF though feed is not required for the effects on hepatic copper levels and on reducing weight, instead bolus dosing is sufficient, but both work.

3. Authors call disulfiram as copper ionophore and claim that it increases the levels of bioavailable copper. Such an effect of disulfiram was, however, never confirmed. Data presented in the manuscript instead indicate the opposite. Despite the accumulation of copper in the liver, circulating levels of copper were decreased after disulfiram. Similarly, ceruloplasmin containing copper was also reduced. Such a claim should be reformulated or supported by more persuasive data.

We disagree with the reviewer, as numerous studies report the use of DSF to enhance bioavailable copper, which is an intracellular measure. Our data clearly show that DSF affects changes in certain cellular tissue types, primarily the liver, and thus redistributes available copper. Holo-Cp (copper bound) is a measure of circulating copper pool in the blood and is not the same as bioavailable copper (which is intracellular available copper pool). Further, the loss of copper from certain copper proteins in the serum (primarily Cp) reflects this ability of DSF to redistribute copper in a biological system. We have cited throughout our manuscript previous studies that have demonstrated that DSF increases copper in specific tissues in vivo. Such as: lines 261-264

As previously mentioned, administration of disulfiram or its metabolite diethyldithiocabamate (DDTC) elevates copper primarily in the liver and to a lesser extent the brain. [30-34].”

4. The authors should at least discuss the previous work demonstrating the anti-obesity effect of disulfiram (Bernier et al., Cell Metabolism, 2020, doi.org/10.1016/j.cmet.2020.04.019), as this paper used very similar methodology, experiments, and readouts, came to a similar conclusion regarding the effect of disulfiram on weight, but without any explanation.

We thank the reviewer for pointing out this recently published work which came out as we were communicating this manuscript. We have now cited this study which independently validated our observations for which we have provided a mechanistic explanation in this study. We have included the following (line 396):

“We previously demonstrated that the copper ionophore clioquinol induces weight loss in several mouse models [61]. Further, disulfiram has been recently described to normalise bodyweight in a mouse model of obesity [78].”

Round 2

Reviewer 1 Report

#Molecules_950209

 The manuscript by Peter M. Meggyesy et al. reports on the application of copper-ionophores to the prevention of obesity. Obesity is currently a risk factor with implication in several diseases, that have a high economic impact. Thus, the identification of therapeutic agents that have the potential to reduce it, without surgical procedures is of outmost importance, due to the eating behaviour of today society. This study has the advantage that are testing a FDA-approved drug for the therapeutic of obesity.

The manuscript is well written and the figures have been adequately chosen.

No further questions or comments.

Nevertheless, the manuscript needs to be uniformly formatted for line spacing and font size and font type.